# Parallel Feature Selection inspired by Group Testing

**Yingbo Zhou**[*]    **Utkarsh Porwal**[*]
CSE Department
SUNY at Buffalo
{yingbozh, utkarshp}@buffalo.edu

**Ce Zhang**
CS Department
University of Wisconsin-Madison
czhang@cs.wisc.edu

**Hung Ngo**
CSE Department
SUNY at Buffalo
hungngo@buffalo.edu

**XuanLong Nguyen**
EECS Department
University of Michigan
xuanlong@umich.edu

**Christopher Ré**
CS Department
Stanford University
chrismre@cs.stanford.edu

**Venu Govindaraju**
CSE Department
SUNY at Buffalo
govind@buffalo.edu

## Abstract

This paper presents a parallel feature selection method for classification that scales up to very high dimensions and large data sizes. Our original method is inspired by group testing theory, under which the feature selection procedure consists of a collection of randomized tests to be performed in parallel. Each test corresponds to a subset of features, for which a scoring function may be applied to measure the relevance of the features in a classification task. We develop a general theory providing sufficient conditions under which true features are guaranteed to be correctly identified. Superior performance of our method is demonstrated on a challenging relation extraction task from a very large data set that have both redundant features and sample size in the order of millions. We present comprehensive comparisons with state-of-the-art feature selection methods on a range of data sets, for which our method exhibits competitive performance in terms of running time and accuracy. Moreover, it also yields substantial speedup when used as a pre-processing step for most other existing methods.

## 1 Introduction

*Feature selection* (FS) is a fundamental and classic problem in machine learning [10, 4, 12]. In classification, FS is the following problem: Given a universe $U$ of possible features, identify a subset of features $F \subseteq U$ such that using the features in $F$ one can build a model to best predict the target class. The set $F$ not only influences the model's accuracy, its computational cost, but also the ability of an analyst to understand the resulting model. In applications, such as gene selection from micro-array data [10, 4], text categorization [3], and finance [22], $U$ may contain hundreds of thousands of features from which one wants to select only a small handful for $F$.

While the overall goal is to have an FS method that is both computationally efficient and statistically sound, natural formulations of the FS problem are known to be NP-hard [2]. For large scale data, scalability is a crucial criterion, because FS often serves not as an end but a means to other sophisticated subsequent learning. In reality, practitioners often resort to heuristic methods, which can broadly be categorized into three types: *wrapper*, *embedded*, and *filter* [10, 4, 12]. In the wrapper method, a classifier is used as a black-box to test on any subset of features. In filter methods no classifier is used; instead, features are selected based on generic statistical properties of the (labeled)

---

[*] * denotes equal contribution

data such as mutual information and entropy. Embedded methods have built in mechanisms for FS as an integral part of the classifier training. Devising a mathematically rigorous framework to explain and justify FS heuristics is an emerging research area. Recently Brown et al. [4] considered common FS heuristics using a formulation based on conditional likelihood maximization.

The primary contribution of this paper is a new framework for *parallelizable* feature selection, which is inspired by the theory of *group testing*. By exploiting parallelism in our test design we obtain a FS method that is easily scalable to millions of features and samples or more, while preserving useful statistical properties in terms of classification accuracy, stability and robustness. Recall that group testing is a combinatorial search paradigm [7] in which one wants to identify a small subset of "positive items" from a large universe of possible items. In the original application, items are blood samples of WWII draftees and an item is *positive* if it is infected with syphilis. Testing individual blood sample is very expensive; the group testing approach is to distribute samples into pools in a smart way. If a pool is tested negative, then all samples in the pool are negative. On the other hand, if a pool is tested positive then at least one sample in the pool is positive. We can think of the FS problem in the group testing framework: there is a presumably small, *unknown* subset $F$ of *relevant features* in a large universe of $N$ features. Both FS and group testing algorithms perform the same basic operation: apply a "test" to a subset $T$ of the underlying universe; this test produces a *score*, $s(T)$, that is designed to measure the quality of the features $T$ (or return positive/negative in the group testing case). From the collection of test scores the relevant features are supposed to be identified. Most existing FS algorithms can be thought of as *sequential* instantiations in this framework[1]: we select the set $T$ to test based on the scores of previous tests. For example, let $\mathbf{X} = (X_1, \ldots, X_N)$ be a collection of features (variables) and $Y$ be the class label. In the *joint mutual information* (JMI) method [25], the feature set $T$ is grown sequentially by adding one feature at each iteration. The next feature's score, $s(X_k)$, is defined relative to the set of features already selected in $T$: $s(X_k) = \sum_{X_j \in T} I(X_k, X_j; Y)$. As each such scoring operation takes a non-negligible amount of time, a sequential method may take a long time to complete.

A key insight is that group testing needs not be done sequentially. With a good pooling design, all the tests can be performed *in parallel* in which we determine the pooling design *without* knowing any pool's test outcome. From the vector of test outcomes, one can identify exactly the collection of positive blood samples. Parallel group testing, commonly called *non-adaptive group testing* (NAGT) is a natural paradigm and has found numerous applications in many areas of mathematics, computer Science, and biology [18]. It is natural to wonder whether a "parallel" FS scheme can be designed for machine learning in the same way NAGT was possible: all feature sets $T$ are specified *in advance*, without knowing the scores of any other tests, and from the final collection of scores the features are identified. This paper initiates a mathematical investigation of this possibility.

At a high level, our *parallel feature selection* (PFS) scheme has three inter-related components: (1) the *test design* indicates the collection of subsets of features to be tested, (2) the *scoring function* $s : 2^{[N]} \to \mathbb{R}$ that assigns a score to each test, and (3) the *feature identification algorithm* that identifies the final selected feature set from the test scores. The design space is thus very large. Every combination of the three components leads to a new PFS scheme.[2] We argue that PFS schemes are preferred over sequential FS for two reasons:

1. *scalability*, the tests in a PFS schem can be performed in parallel, and thus the scheme can be scaled to large datasets using standard parallel computing techniques, and
2. *stability*, errors in individual trials do not affect PFS methods as dramatically as sequential methods. In fact, we will show in this paper that increasing the number of tests improves the accuracy of our PFS scheme.

We propose and study one such PFS approach. We show that our approach has comparable (and sometimes better) empirical quality compared to previous heuristic approaches while providing sound statistical guarantees and substantially improved scalability.

**Our technical contributions**   We propose a simple approach for the first and the third components of a PFS scheme. For the second component, we prove a sufficient condition on the scoring function under which the feature identification algorithm we propose is guaranteed to identify *exactly* the set

of original (true) features. In particular, we introduce a notion called *C-separability*, which roughly indicates the strength of the scoring function in separating a relevant feature from an irrelevant feature. We show that when $s$ is $C$-separable and we can estimate $s$, we are able to guarantee exact recovery of the right set of features with high probability. Moreover, when $C > 0$, the number of tests can be asymptotically logarithmic in the number of features in $U$.

In theory, we provide sufficient conditions (a Naïve Bayes assumption) according to which one can obtain separable scoring functions, including the KL divergence and mutual information (MI). In practice, we demonstrate that MI is separable even when the sufficient condition does not hold, and moreover, on generated synthetic data sets, our method is shown recover *exactly* the relevant features. We proceed to provide a comprehensive evaluation of our method on a range of real-world data sets of both large and small sizes. It is the large scale data sets where our method exhibits superior performance. In particular, for a huge relation extraction data set (TAC-KBP) that has millions redundant features and samples, we outperform all existing methods in accuracy and time, in addition to generating plausible features (in fact, many competing methods could not finish the execution). For the more familiar NIPS 2013 FS Challenge data, our method is also competitive (best or second-best) on the two largest data sets. Since our method hinges on the accuracy of score functions, which is difficult achieve for small data, our performance is more modest in this regime (staying in the middle of the pack in terms of classification accuracy). Nonetheless, we show that our method can be used as a preprocessing step for other FS methods to eliminate a large portion of the feature space, thereby providing substantial computational speedups while retaining the accuracy of those methods.

## 2 Parallel Feature Selection

**The general setting**     Let $N$ be the total number of input features. For each subset $T \subseteq [N] := \{1, \ldots, N\}$, there is a *score* $s(T)$ normalized to be in $[0, 1]$ that assesses the "quality" of features in $T$. We select a collection of $t$ *tests*, each of which is a subset $T \subseteq [N]$ such that from the scores of all tests we can identify the *unknown* subset $F$ of $d$ relevant variables that are most important to the classification task. We encode the collection of $t$ tests with a binary matrix $\mathbf{A} = (a_{ij})$ of dimension $t \times N$, where $a_{ij} = 1$ iff feature $j$ belongs to test $i$. Corresponding to each row $i$ of $\mathbf{A}$ is a "test score" $s_i = s(\{j \mid a_{ij} = 1\}) \in [0, 1]$. Specifying $\mathbf{A}$ is called *test design*, identifying $F$ from the score vector $(s_i)_{i \in [t]}$ is the job of the *feature identification algorithm*. The scheme is inherently parallel because *all* the tests must be specified in advance and executed in parallel; then the features are selected from all the test outcomes.

**Test design and feature identification**     Our test design and feature identification algorithms are *extremely simple*. We construct the test matrix $\mathbf{A}$ randomly by putting a feature in the test with probability $p$ (to be chosen later). Then, from the test scores we *rank* the features and select $d$ top-ranked features. The ranking function is defined as follows. Given a $t \times N$ test matrix $\mathbf{A}$, let $\mathbf{a}^j$ denote its $j$th column. The dot-product $\langle \mathbf{a}^j, \mathbf{s} \rangle$ is the total score of all the tests that feature $j$ participates in. We define $\rho(j) = \langle \mathbf{a}^j, \mathbf{s} \rangle$ to be the *rank* of feature $j$ with respect to the test matrix $\mathbf{A}$ and the score function $s$.

**The scoring function**     The crucial piece stiching together the entire scheme is the scoring function. The following theorem explains why the above test design and feature identification strategy make sense, as long as one can choose a scoring function $s$ that satisfies a natural *separability* property. Intuitively, separable scoring functions require that adding more hidden features into a test set increase its score.

**Definition 2.1** (Separable scoring function)**.** Let $C \geq 0$ be a real number. The score function $s : 2^{[N]} \to [0, 1]$ is said to be *C-separable* if the following property holds: for every $f \in F$ and $\tilde{f} \notin F$, and for every $T \subseteq [N] - \{f, \tilde{f}\}$, we have $s(T \cup \{f\}) - s(T \cup \{\tilde{f}\}) \geq C$.

In words, with a separable scoring function adding a relevant feature should be better than adding an irrelevant feature to a given subset $T$ of features. Due to space limination, the proofs of the following theorem, propositions, and corollaries can be found in the supplementary materials. The essence of the idea is that, when $s$ can separate relevant features from irrelevant features, with high probability a relevant feature will be ranked higher than an irrelevant feature. Hoeffding's inequality is then used to bound the number of tests.

**Theorem 2.2.** *Let* $\mathbf{A}$ *be the random* $t \times N$ *test matrix obtained by setting each entry to be* 1 *with probability* $p \in [0, 1]$ *and* 0 *with probability* $1 - p$. *If the scoring function* $s$ *is* $C$-*separable, then the expected rank of a feature in* $F$ *is at least the expected rank of a feature not in* $F$.

*Furthermore, if* $C > 0$, *then for any* $\delta \in (0, 1)$, *with probability at least* $1 - \delta$ *every* feature in $F$ *has rank higher than* every *feature not in* $F$, *provided that the number of tests* $t$ *satisfies*

$$t \geq \frac{2}{C^2 p^2 (1-p)^2} \log \left( \frac{d(N-d)}{\delta} \right). \tag{1}$$

By setting $p = 1/2$ in the above theorem, we obtain the following. It is quite remarkable that, assuming we can estimate the scores accurately, we only need about $O(\log N)$ tests to identify $F$.

**Corollary 2.3.** *Let* $C > 0$ *be a constant such that there is a* $C$-*separable scoring function* $s$. *Let* $d = |F|$, *where* $F$ *is the set of hidden features. Let* $\delta \in (0, 1)$ *be an arbitrary constant. Then, there is a distribution of* $t \times N$ *test matrices* $\mathbf{A}$ *with* $t = O(\log(d(N-d)/\delta))$ *such that, by selecting a test matrix randomly from the distribution, the* $d$ *top-ranked features are* exactly *the hidden features with probability at least* $1 - \delta$.

Of course, in reality estimating the scores accurately is a very difficult problem, both statistically and computationally, depending on what the scoring function is. We elaborate more on this point below. But first, we show that separable scoring functions exist, under certain assumption about the underlying distribution.

**Sufficient conditions for separable scoring functions** We demonstrate the existence of separable scoring functions given some sufficient conditions on the data. In practice, loss functions such as classification error and other surrogate losses may be used as scoring functions. For binary classification, information-theoretic quantities such as Kullback-Leibler divergence, Hellinger distance and the total variation — all of which special cases of $f$-divergences [5, 1] — may also be considered. For multi-class classification, mutual information (**MI**) is a popular choice.

The data pairs $(\mathbf{X}, Y)$ are assumed to be iid samples from a joint distribution $P(\mathbf{X}, Y)$. The following result shows that under the so-called "naive Bayes" condition, i.e., all components of random vector $\mathbf{X}$ are conditionally independent given label variable $Y$, the Kullback-Leibler distance is a separable scoring function in a binary classification setting:

**Proposition 2.4.** *Consider the binary classification setting, i.e.,* $Y \in \{0, 1\}$ *and assume that the naive Bayes condition holds. Define score function to be the Kullback-Leibler divergence:*

$$s(T) := KL(P(\mathbf{X}_T | Y = 0) || P(\mathbf{X}_T | Y = 1)).$$

*Then* $s$ *is a separable scoring function. Moreover,* $s$ *is* $C$-*separable, where* $C := \min_{f \in F} s(f)$.

**Proposition 2.5.** *Consider the multi-class classification setting, and assume that the naive Bayes condition holds. Moreover, for any pair* $f \in F$ *and* $\tilde{f} \notin F$, *the following holds for any* $T \subseteq [N] - \{f, \tilde{f}\}$

$$I(X_f; Y) - I(X_f; \mathbf{X}_T) \geq I(X_{\tilde{f}}; Y) - I(X_{\tilde{f}}; \mathbf{X}_T).$$

*Then, the MI function* $s(T) := I(\mathbf{X}_T; Y)$ *is a separable scoring function.*

We note the naturalness of the condition so required, as quantity $I(X_f; Y) - I(X_f; X_T)$ may be viewed as the relevance of feature $f$ with respect to the label $Y$, subtracted by the redundancy with other existing features $T$. If we assume further that $\mathbf{X}_{\tilde{f}}$ is independent of both $\mathbf{X}_T$ and the label $Y$, and there is a positive constant $C$ such that $I(X_f; Y) - I(X_f; X_T) \geq C$ for any $f \in F$, then $s(T)$ is obviously a $C$-separable scoring function. It should be noted that *the naive Bayes conditions are sufficient, but not necessary* for a scoring function to be $C$-separable.

**Separable scoring functions for filters and wrappers.** In practice, information-based scoring functions need to be estimated from the data. Consistent estimators of scoring functions such as KL divergence (more generally $f$-divergences) and MI are available (e.g., [20]). This provides the theoretical support for applying our test technique to filter methods: when the number of training data is sufficiently large, a consistent estimate of a separable scoring function must also be a separable scoring function. On the other hand, a wrapper method uses a classification algorithm's performance as a scoring function for testing. Therefore, the choice of the underlying (surrogate) loss function plays a critical role. The following result provides the existence of loss functions which induce separable scoring functions for the wrapper method:

**Proposition 2.6.** *Consider the binary classification setting, and let $P_0^T := P(\boldsymbol{X}_T|Y = 0)$, $P_1^T := P(\boldsymbol{X}_T|Y = 1)$. Assume that an f-divergence of the form: $s(T) = \int \phi(dP_0^T/dP_1^T)dP_1^T$ is a separable scoring function for some convex function $\phi : \mathbb{R}_+ \to \mathbb{R}$. Then there exists a surrogate loss function $l : \mathbb{R} \times \mathbb{R} \to \mathbb{R}_+$ under which the minimum l-risk: $R_l(T) := \inf_g \mathbb{E}\left[l(Y, g(\boldsymbol{X}_T))\right]$ is also a separable scoring function. Here the infimum is taken over all measurable classifier functions $g$ acting on feature input $\boldsymbol{X}_T$, $\mathbb{E}$ denotes expectation with respect to the joint distribution of $\boldsymbol{X}_T$ and $Y$.*

This result follows from Theorem 1 of [19], who established a precise correspondence between $f$-divergences defined by convex $\phi$ and equivalent classes of surrogate losses $l$. As a consequence, if the Hellinger distance between $P_0^T$ and $P_1^T$ is separable, then the wrapper method using the Adaboost classifier corresponds to a separable scoring function. Similarly, a separable Kullback-Leibler divergence implies that of a logistic regression based wrapper; while a separable variational distance implies that of a SVM based wrapper.

## 3 Experimental results

### 3.1 Synthetic experiments

In this section, we synthetically illustrate that separable scoring functions exist and our PFS framework is sound beyond the Naïve Bayes assumption (**NBA**). We first show that MI is $C$-separable for large $C$ even when the NBA is violated. The NBA was only needed in Propositions 2.4 and 2.5 in order for the proofs to go through. Then, we show that our framework recovers *exactly* the relevant features for two common classes of input distributions.

We generate $1,000$ data points from two separated 2-D Gaussians with the same covariance matrix but different means, one centered at $(-2,-2)$ and the other at $(2,2)$. We start with the identity covariance matrix, and gradually change the off diagonal element to $-0.999$, representing highly correlated features. Then, we add 1,000 dimensional zero mean Gaussian noise with the same covariance matrix, where the diagonal is 1 and the off-diagonal elements increases from 0 gradually to 0.999. We then calculate the MI between two features and the class label, and the two features are selected in three settings: 1) the two genuine dimensions; 2) one of the genuine feature and one from the noisy dimensions; 3) two random pair from the noisy dimensions. The MI that we get from

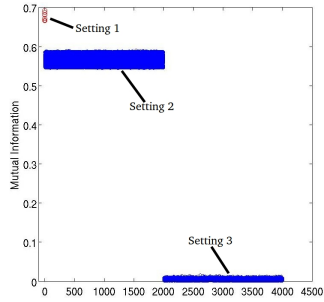

Figure 1: Illustration of MI as a separable scoring function for the case of statistically dependent features. The top left point shows the scores for the 1st setting; the middle points shows the scores for the 2nd setting; and the bottom points shows the scores for the 3rd setting.

these three conditions is shown in Figure 1. It is clear from this figure MI is a separable scoring function, despite the fact that the NBA is violated.

We also synthetically evaluated our entire PFS idea, using two multinomials and two Gaussians to generate two binary classification task data. Our PFS scheme is able to capture *exactly* the relevant features in most cases. Details are in the supplementary material section due to lack of space.

### 3.2 Real-world data experiment results

This section evaluates our approach in terms of accuracy, scalability, and robustness accross a range of real-world data sets: small, medium, and large. We will show that our PFS scheme works very well on medium and large data sets; because, as was shown in Section 3.1, with sufficient data to estimate test scores, we expect our method to work well in terms of accuracy. On the small datasets, our approach is only competitive and does not dominate existing approaches, due to the lack of data to estimate scores well. However, we show that we can still use our PFS scheme as a pre-processing step to filter down the number of dimensions; this step reduces the dimensionality, helps speed up existing FS methods from 3-5 times while keeps their accuracies.

#### 3.2.1 The data sets and competing methods

**Large:** TAC-KBP is a large data set with the number of samples and dimensions in the millions[3]; its domain is on relation extraction from natural language text. **Medium:** GISETTE and MADE-

LON are two largest data sets from the NIPS 2003 feature selection challenge[4], with the number of dimensions in the thousands. **Small:** Colon, Leukemia, Lymph, NCI9, and Lung are chosen from the small Micro-array datasets [6], along with the UCI datasets[5]. These sets typically have a few hundreds to a few thousands variables, with only tens of data samples.

We compared our method with various baseline methods including mutual information maximization[14] (MIM), maximum relevancy minimum redundancy[21] (MRMR), conditional mutual information maximization[9] (CMIM), joint mutual information[25] (JMI), double input symmetrical relevance[16] (DISR), conditional infomax feature extraction[15] (CIFE), interaction capping[11] (ICAP), fast correlation based filter[26] (FCBF), local learning based feature selection [23] (LOGO), and feature generating machine [24] (FGM).

### 3.2.2 Accuracy

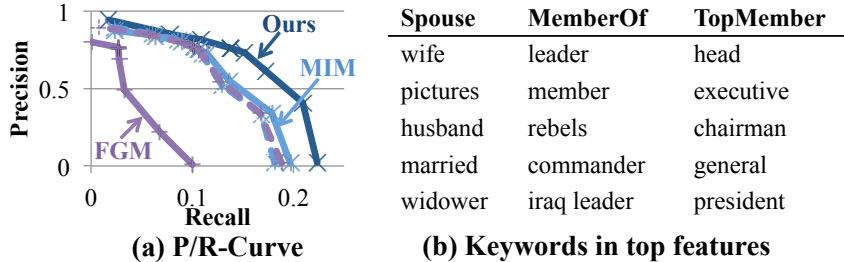

| Spouse | MemberOf | TopMember |
|--------|----------|-----------|
| wife | leader | head |
| pictures | member | executive |
| husband | rebels | chairman |
| married | commander | general |
| widower | iraq leader | president |

**(a) P/R-Curve** **(b) Keywords in top features**

Figure 2: Result from different methods on TAC-KBP dataset. (a) Precision/Recall of different methods; (b) Top-5 keywords appearing in the Top-20 features selected by our method. Dotted lines in (a) are FGM (or MIM) with our approach as pre-processing step.

**Accuracy results on large data set.** As shown in Figure 2(a), our method dominates both MIM and FGM. Given the same precision, our method achieves 2-14× higher recall than FGM, and 1.2-2.4× higher recall than MIM. Other competitors do not finish execution in 12 hours. We compare the top-features produced by our method and MIM, and find that our method is able to extract features that are strong indicators only when they are combined with other features, while MIM, which tests features individually, ignores this type of combination. We then validate that the features selected by our method makes intuitive sense. For each relation, we select the top-20 features and report the keyword in these features.[6] As shown in Figure 2(b), these top-features selected by our method are good indicators of each relation. We also observe that using our approach as the pre-processing step improves the quality of FGM significantly. In Figure 2(a) (the broken lines), we run FGM (MIM) on the top-10K features produced by our approach. We see that running FGM with pre-processing achieves up to 10× higher recall given the same precision than running FGM on all 1M features.

**Accuracy results on medium data sets** Since the focus of the evaluation is to analyze the efficacy of feature selection approaches, we employed the same strategy as Brown et al.[4] *i.e.* the final classification is done using $k$-nearest neighbor classifier with $k$ fixed to three, and applied Euclidean distance[7].

We denote our method by $F_k$ (and $W_k$), where F denotes filter (and W denotes wrapper method). $k$ denotes the number of tests (*i.e.* let $N$ be the dimension of data, then the total number of tests is $kN$). We bin each dimension of the data into five equal distanced bins when the data is real valued, otherwise the data is not processed[8]. MI is used as the scoring function for filter method, and log-likelihood is used for scoring the wrapper method. The wrapper we used is logistic regression[9].

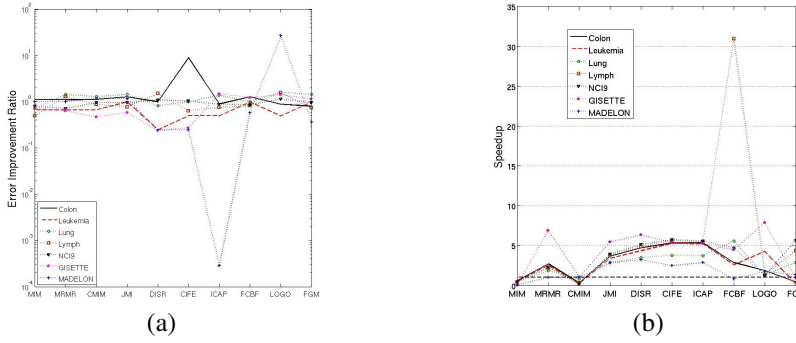

(a)                                    (b)

Figure 3: Result from real world datasets: a) curve showing the ratio between the errors of various methods applied on original data and on filtered data, where a large portion of the dimension is filtered out (value larger than one indicates performance improvement); b) the speed up we get by applying our method as a pre-processing method on various methods across different datasets, the flat dashed line indicates the location where the speed up is one.

For GISETTE we select up to 500 features and for MADELON we select up to 100 features. To get the test results, we use the features according to the smallest validation error for each method, and the results on test set are illustrated in table 4.

Table 1: Test set balanced error rate (%) from different methods on NIPS datasets

| Datasets | Best Perf. | 2nd Best Perf. | 3rd Best Perf. | Median Perf. | Ours ($F_3$) | Ours ($W_3$) | Ours ($F_{10}$) | Ours ($W_{10}$) |
|---|---|---|---|---|---|---|---|---|
| GISETTE | **2.15** | 3.06 | 3.09 | 3.86 | 4.85 | 2.72 | 4.69 | 2.89 |
| MADELON | 10.61 | 11.28 | 12.33 | 25.92 | 22.61 | **10.17** | 18.39 | **10.50** |

**Accuracy results on the small data sets.**   As expected, due to the lack of data to estimate scores, our accuracy performance is average for this data set. Numbers can be found in the supplementary materials. However, as suggested by theorem A.3 (in supplementary materials), our method can also be used as a preprocessing step for other feature selection method to eliminate a large portion of the features. In this case, we use the filter methods to filter out $e + 0.1$ of the input features, where $e$ is the desired proportion of the features that one wants to reserve.

Using our method as preprocessing step achieves 3-5 times speedup as compare to the time spend by original methods that take multiple passes through the datasets, and keeps or improves the performance in most of the cases (see figure 3 a and b). The actual running time can be found in supplementary materials.

### 3.2.3   Scalability

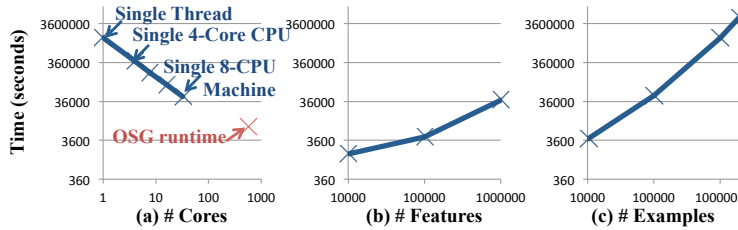

Figure 4: Scalability Experiment of Our Approach

We validate that our method is able to run on large-scale data set efficiently, and the ability to take advantage of parallelism is the key to its scalability.

**Experiment Setup**    Given the TAC-KBP data set, we report the execution time by varying the degree of parallelism, number of features, and number of examples. We first produce a series of data sets by sub-sampling the original data set with different number examples ($\{10^4, 10^5, 10^6\}$) and number of features ($\{10^4, 10^5, 10^6\}$). We also try different degree of parallelism by running our approach using a single thread, 4-threads on a 4-core CPU, 32 threads on a single 8-CPU (4-core/CPU) machine, and multiple machines available in the national Open Science Grid (OSG). For each combination of number of features, number of examples, and degree of parallelism, we estimate the throughput as the number of tests that we can run in 1 second, and estimate the total running time accordingly. We also ran our largest data set ($10^6$ rows and $10^6$ columns) on OSG and report the actual run time.

**Degree of Parallelism**    Figure 4(a) reports the (estimated) run time on the largest data set ($10^6$ rows and $10^6$ columns) with different degree of parallelism. We first observe that running our approach requires non-trivial amount of computational resources–if we only use a single thread, we need about 400 hours to finish our approach. However, the running time of our approach decreases linearly with the number of cores that we used. If we run our approach on a single machine with 32 cores, it finishes in just 11 hours. This linear speed-up behavior allows our approach to scale to very large data set–when we run our approach on the national Open Science Grid, we observed that our approach is able to finish in 2.2 hours (0.7 hours for actual execution, and 1.5 hours for scheduling overhead).

**The Impact of Number of Features and Number of Examples**    Figure 4(b,c) report the run time with different number of features and number of examples, respectively. In Figure 4(b), we fix the number of examples to be $10^5$, and vary the number of features, and in Figure 4(c), we fix the number of features to be $10^6$ and vary the number of examples. We see that as the number of features or the number of examples increase, our approach uses more time; however, the running time never grows super-linearly. This behavior implies the potential of our approach to scale to even larger data sets.

### 3.2.4    Stability and robustness

Our method exhibits several robustness properties. In particular, the proof of Theorem 2.2 suggests that as the number of tests are increased the performance also improves. Therefore, in this section we empirically evaluate this observation. We picked four datasets: KRVSKP, Landset, Splice and Waveform from the UCI datasets and both NIPS datasets.

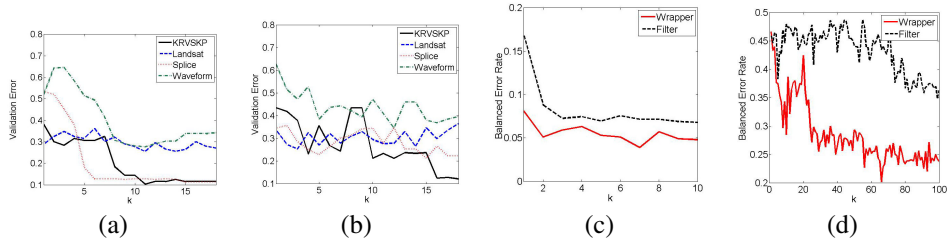

Figure 5: Change of performance with respect of number of tests on several UCI datasets with (a) filter and (b) wrapper methods; and (c) GISETTE and (d) MADELON datasets.

The trend is pretty clear as can be observed from figure 5. The performance of both wrapper and filter methods improves as we increase the number of tests, which can be attributed to the increase of robustness against inferior estimates for the test scores as the number of tests increases. In addition, apart from MADELON dataset, the performance converges fast, normally around $k = 10 \sim 15$.

Additional stability experiments can be found in the supplementary materials, where we evaluate ours and other methods in terms of consistency index.

## Footnotes

[1]A notable exception is the MIM method, which is easily parallelizable and can be regarded as a special implementation of our framework

[2]It is important to emphasize that this PFS framework is applicable to both filter and wrapper approaches. In the wrapper approach, the score $s(T)$ might be the training error of some classifier, for instance.

[3] http://nlp.cs.qc.cuny.edu/kbp/2010/

[4] http://www.nipsfsc.ecs.soton.ac.uk/datasets/

[5] http://archive.ics.uci.edu/ml/

[6] Following the syntax used by Mintz et al. [17], if a feature has the form $[\Uparrow_{poss} \, wife \, \Downarrow_{prop\_of}]$, we report the keyword as *wife* in Figure 2(b).

[7] The classifier for FGM is linear support vector machine (SVM), since it optimized for the SVM criteria.

[8] For SVM based method, the real valued data is not processed, and all data is normalized to have unit length.

[9] The logistic regressor used in wrapper is only to get the testing scores, the final classification scheme is still $k$-NN.

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
