[Supplementary Material · supplementary.pdf]

## Supplementary Material

## A   Missing proofs from Section 2

### A.1   Proof of Theorem 2.2

*Proof.* Fix a feature $f \in F$ and a feature $\tilde{f} \notin F$. Recall we used $\mathbf{a}^j$ to denote the $j$th column of the test matrix $\mathbf{A}$. For each row $i \in [t]$, define the random variable $X_i := a_{if} s_i - a_{i\tilde{f}} s_i$, which is the contribution of the $i$th test to the difference $\rho(f) - \rho(\tilde{f})$. In particular,

$$\rho(f) - \rho(\tilde{f}) = \langle \mathbf{a}^f, \mathbf{s} \rangle - \langle \mathbf{a}^{\tilde{f}}, \mathbf{s} \rangle = \sum_{i=1}^{t} X_i.$$

The variables $X_i$ are identically and independently distributed. We first estimate $\mathbb{E}[X_i]$. Let $T_i$ denote the $i$th test, i.e. $T_i = \{ j \mid a_{ij} = 1 \}$. Then, it is easy to see that $\mathbb{E}[X_i \mid \text{both } f, \tilde{f} \in T_i] = 0$, and $\mathbb{E}[X_i \mid \text{both } f, \tilde{f} \notin T_i] = 0$. Thus, letting $q = 1 - p$ to shorten the notations, we have

$$\mathbb{E}[X_i]$$
$$= \mathbb{E}[X_i \mid f \in T_i, \tilde{f} \notin T_i] \cdot \mathbb{P}[f \in T_i, \tilde{f} \notin T_i] +$$
$$\quad \mathbb{E}[X_i \mid f \notin T_i, \tilde{f} \in T_i] \cdot \mathbb{P}[f \notin T_i, \tilde{f} \in T_i]$$
$$= pq\mathbb{E}[X_i \mid f \in T_i, \tilde{f} \notin T_i] + pq\mathbb{E}[X_i \mid f \notin T_i, \tilde{f} \in T_i]$$
$$= pq \left( \sum_{T \subseteq [N] - \{f, \tilde{f}\}} s(T_i) \cdot \mathbb{P}[T_i = T \cup \{f\} \mid f \in T_i, \tilde{f} \notin T_i] \right)$$
$$\quad -pq \left( \sum_{T \subseteq [N] - \{f, \tilde{f}\}} s(T_i) \cdot \mathbb{P}[T_i = T \cup \{\tilde{f}\} \mid f \notin T_i, \tilde{f} \in T_i] \right)$$
$$= pq \left( \sum_{T \subseteq [N] - \{f, \tilde{f}\}} s(T \cup \{f\}) \cdot p^{|T|} q^{N-2-|T|} \right)$$
$$\quad -pq \left( \sum_{T \subseteq [N] - \{f, \tilde{f}\}} s(T \cup \{\tilde{f}\}) \cdot p^{|T|} q^{N-2-|T|} \right)$$
$$= pq \left( \sum_{T \subseteq [N] - \{f, \tilde{f}\}} \left( s(T \cup \{f\}) - s(T \cup \{\tilde{f}\}) \right) \cdot p^{|T|} q^{N-2-|T|} \right)$$
$$\geq Cpq \left( \sum_{T \subseteq F - \{f, \tilde{f}\}} p^{|T|} q^{N-2-|T|} \right) = Cpq.$$

Consequently, when $C \geq 0$ every term in the summation above is non-negative, implying that $\mathbb{E}[X_i] \geq 0$, which in turn implies $\mathbb{E}[\rho(f)] \geq \mathbb{E}[\rho(\tilde{f})]$. Since the $X_i$ are i.i.d. in $[-1, 1]$, by Hoeffding's inequality [8], when $C > 0$ we have

$$\mathbb{P}[\rho(f) - \rho(\tilde{f}) \leq 0] \leq \exp \left\{ \frac{-2t^2 C^2 p^2 q^2}{4t} \right\}.$$

The probability that there is *some* pair $f \in F, \tilde{f} \notin F$ for which $\rho(f) - \rho(\tilde{f}) \leq 0$ is thus at most $d(N - d) \exp \left\{ \frac{-t C^2 p^2 q^2}{2} \right\}$. The last expression is at most $\delta$ when $t$ satisfies (1). □

### A.2   Proof of Proposition 2.4 and Proposition 2.5

A special case of separable scoring function is a scoring function satisfying a monotonicity condition: if a subset $T_2$ of features has more relevant features than another subset $T_1$, then $s(T_2)$ has to be better than $s(T_1)$.

**Definition A.1** (Monotone scoring function)**.** Let $C \geq 0$ be a real number. The score function $s : 2^{[N]} \to [0, 1]$ is said to be $C$-*monotone* if the following property holds: for any two subsets $T_1, T_2 \subseteq [N]$ such that $T_1 \cap F$ is a proper subset of $T_2 \cap F$, we have $s(T_2) - s(T_1) \geq C$. The 0-monotone scoring functions are called *monotone* for short.

**Proposition A.2.** *If $s$ is a $C$-monotone scoring function, then it is a $C$-separable scoring function.*

*Proof.* Fix $f \in F$, $\tilde{f} \notin F$, and $T \subset [N] - \{f, \tilde{f}\}$. Let $T_2 = T \cup \{f\}$ and $T_1 = T \cup \{\tilde{f}\}$. Then, $T_1 \cap F \subset T_2 \cap F$. Hence, $s(T_2) - s(T_1) \geq C$, as desired. □

From the above proposition, to show that a function is separable it is sufficient to show that it is monotone.

*Proof of Proposition 2.4.* Due to conditional independence, it can be checked that $s(T) = \sum_{f \in T} s(f)$. From this the claim can be easily verified. □

*Proof of Proposition 2.5.* Due to a basic property of mutual information, $s(T \cup \{f\}) = I(\boldsymbol{X}_{T \cup \{f\}}; Y) = H(\boldsymbol{X}_{T \cup \{f\}}) - H(\boldsymbol{X}_{T \cup \{f\}}|Y) = H(\boldsymbol{X}_{T \cup \{f\}}) - H(\boldsymbol{X}_T|Y) - H(X_f|Y)$, where the last identity is due to the conditional independence assumption. Fix $f \in F$ and $\tilde{f} \notin F$. Since $H(\boldsymbol{X}_{T \cup \{f\}}) = H(\boldsymbol{X}_T) + H(X_f) - I(\boldsymbol{X}_T; X_f)$, we have $s(T \cup \{f\}) = I(\boldsymbol{X}_T; Y) + I(X_f; Y) - I(\boldsymbol{X}_T; X_f)$. Combine with a similar formulae for $s(T \cup \{\tilde{f}\})$, we obtain:

$$s(T \cup \{f\}) - s(T \cup \{\tilde{f}\}) = (I(X_f; Y) - I(\boldsymbol{X}_T; X_f)) - (I(X_{\tilde{f}}; Y) - I(\boldsymbol{X}_T; X_{\tilde{f}})) \geq 0,$$

which concludes the proof. □

## A.3 On eliminating irrelevant features

The rank $\rho(f)$ of a feature is proportional to the average score of all tests that the feature $f$ participates in. If $f$ is "lucky" enough to participate in tests that contain relevant features, its rank might be inflated. This observation leads to our second idea: we need a way to quickly eliminate features that are likely to be irrelevant.

**Theorem A.3.** *Let $F$ be the set of hidden relevant features. Let $d = |F|$. Let $\mathbf{A}$ be the random $t \times N$ test matrix obtained by setting each entry to be 1 with probability $p \in [0, 1]$ and 0 with probability $1 - p$. For an irrelevant feature $\tilde{f} \notin F$, let $U_{\tilde{f}}$ denote the total number of tests that $\tilde{f}$ belongs, and $V_{\tilde{f}}$ the total number of tests that $\tilde{f}$ belongs but none of the relevant features belong.*

*For any $\delta \in (0, 1)$, and any $\beta$ such that $0 < \beta < (1-p)^d$, the following holds:*

$$\mathbb{P}[V_{\tilde{f}} \geq \beta U_{\tilde{f}} \text{ for all } \tilde{f} \notin F] \geq 1 - \delta,$$

*provided that the total number of tests is at least*

$$t \geq \frac{1}{2} \cdot \frac{(1+\beta)^2}{p^2((1-p)^d - \beta)^2} \log((N-d)/\delta). \tag{2}$$

*Proof.* Let $\tilde{f}$ be an arbitrary irrelevant feature. For each $j \in [t]$, let $X_j$ be the indicator variable for the event that $\tilde{f}$ is in test $j$, and $Y_j$ be the indicator variable for the event that $\tilde{f}$ belongs to the $j$th test but none of the relevant features are in test $j$. Then, $U_{\tilde{f}} = \sum_{j \in [t]} X_j$ and $V_{\tilde{f}} = \sum_{j \in [t]} Y_j$. It follows that $\mathbb{E}[Y_j - \beta X_j] = p(1-p)^d - \beta p$. Furthermore, we have $Y_j - \beta X_j \in [-\beta, 1]$, and for $j \in [t]$ the variables $Y_j - \beta X_j$ are independent. Hence, by Hoeffding bound we have

$$\begin{aligned} \mathbb{P}[V_{\tilde{f}} < \beta U_{\tilde{f}}] &= \mathbb{P}[\sum_{j \in [t]} (Y_j - \beta X_j) < 0] \\ &\leq \exp\left\{ \frac{-2t^2 p^2((1-p)^d - \beta)^2}{t(1+\beta)^2} \right\} \\ &= \exp\left\{ \frac{-2t p^2((1-p)^d - \beta)^2}{(1+\beta)^2} \right\}. \end{aligned}$$

Hence, due to condition 2,

$$\mathbb{P}[V_{\tilde{f}} < \beta U_{\tilde{f}} \text{ for some } \tilde{f} \notin F] \leq (N-d) \exp\left\{ \frac{-2t p^2((1-p)^d - \beta)^2}{(1+\beta)^2} \right\} \leq \delta.$$

□

Figure 6: Box plot from synthetic data on a) the identifiability of original features with abundant data from multinomial distribution, b) the identifiability of original features with reasonable-sized data from multinomial distribution, c) the identifiability of original features with abundant data from discretized Gaussian distribution, d) the identifiability of original features with reasonable-sized data from discretized Gaussian distribution.

The above theorem is useful when we can find a score function such that the tests that contain **no** relevant have low scores, say less than some threshold $\theta$. In that case, the natural algorithm is to first eliminate all features such that at least a $\beta$ fraction of its tests score lower than $\theta$.

To make use of the above algorithm, we need to set the parameters. For example, suppose we set $p = 1/d$. Then $(1 - p)^d = (1 - 1/d)^d$ is an increasing function in $d$ that tends to $1/e \approx 0.37$ fairly quickly. Hence, $d \geq 4$ we can pick $\beta = 0.25$ (or more). But there is a tradeoff between $\beta$ and the number of tests $t$, hence we do not want to pick $\beta$ to be too close to $(1 - p)^d$.

As a second example, suppose we set $p = 1/(2d)$. Then, $(1 - p)^d = (1 - 1/\sqrt{d})^d \rightarrow 1/\sqrt{e} \approx 0.61$. In this case we can even pick $\beta = 1/2$.

# B    Additional details on synthetic experiment results

We evaluate the entire PFS idea synthetically. We generate a simple categorical binary class dataset using two multinomial distributions. Let $N_o$ be the number of original data dimensions (*i.e.* where the data is actually dependent on). The $N_n$ noisy dimensions are generated with uniform probability for all $N_n$ dimensions, so the synthetic data generated is of dimension $N = N_o + N_n$. The number of trials were restricted to five in our data generation. As theorem 2.2 suggests, by setting $p = 0.5$ we only need logarithmic number of tests with respect to the number of feature dimensions, but it will lead to inaccurate score estimation when $N$ is large and the number of data samples are small. Therefore, we first simulate a case where we have abundant samples by sampling $10,000$ samples for each class with $N_o \in \{2, 4, 10\}$ and $N_n \in \{10, 20, 30\}$. We set $p = 0.5$ and $t = \lceil \frac{2}{p^2(1-p)^2} \log(N_o N_n / \delta) \rceil$ where $\delta = 0.01$. In addition, to attain a more realistic setting, we generate $1,000$ samples for each class, with $N_o \in \{4, 10, 50, 100\}$ and $N_n \in \{10, 50, 100, 500\}$. We set $p = \frac{3}{N}$ so that we can get reasonable score estimate and $t = 10N$. To account for the randomness of the test, we ran every experiment 100 times; the result is shown in Figure 6(a) and (b) respectively. It is clear that most of the time all the original dimensions are contained in the top $D_o$ ranked features, in particular, when the score can be estimated reasonably well, the top $D_o$ features contains exactly all the original features (see figure 6 (a) and (c)).

Since not all real world data are categorical valued, we simulated another real-valued binary dataset. The $N_o$ original data dimensions are generated from two Gaussian with mean 3 and $-3$. They share the same variance, and it is uniformly sampled from the interval $(0, 1]$. The $N_n$ noisy dimensions are generated from Gaussian with mean sample uniformly from interval $[-1, 1]$ and variance uniformly sampled from interval $(0, 1]$. We then quantize each dimension into five equal distanced bins. We use the exactly same settings as the previous experiments, and the result is illustrated in figure 6 (c) and (d), it can be observed that the performance are consistent with the last set of experiments.

# C    Additional results on the small and medium data sets

## C.1    Accuracy and runtime results on Micro-array dataset

The Colon and Leukemia dataset are both binary class dataset that contains 62 samples with 2,000 dimensions and 72 data points with 7,070 dimensions respectively; the Lymph and NCI9 dataset both have 9 classes and

respectively contain 96 samples with 4,026 dimensions and 60 samples with 9,712 dimensions; The Lung dataset contains 73 data samples of 325 dimensions and is a 7-class dataset.

We set the maximum number of selected features to be 50. $d$ Models were trained for each dataset with the top $d$ features where $d$ varies from 1 to 50, and we report the best overall *leave-one-out* classification error among all 50 combinations of features. For the wrapper method we set $p = 10/N$ and for filter method we set $p = 4/N$, where $N$ is the dimension of data.

Table 2: Leave one out error on micro-array datasets from various methods

| Method/Dataset | Colon | Leukemia | Lung | Lymph | NCI9 |
|---|---|---|---|---|---|
| MIM | 10 | 2 | 14 | 13 | 25 |
| MIM (Filtered) | 10 | 2 | 13 | 13 | 25 |
| MRMR | 9 | 2 | 13 | 7 | 23 |
| MRMR (Filtered) | 9 | 2 | 13 | 7 | 23 |
| CMIM | 9 | 1 | 9 | 9 | 26 |
| CMIM (Filtered) | 9 | 1 | 9 | 9 | 26 |
| JMI | 9 | 1 | 11 | 9 | 24 |
| JMI (Filtered) | 9 | 1 | 11 | 9 | 24 |
| DISR | 8 | 1 | 13 | 11 | 24 |
| DISR (Filtered) | 8 | 1 | 13 | 11 | 24 |
| CIFE | 9 | 3 | 19 | 26 | 31 |
| CIFE (Filtered) | 9 | 3 | 9 | 10 | 35 |
| ICAP | 8 | 3 | 10 | 8 | 24 |
| ICAP (Filtered) | 7 | 2 | 9 | 9 | 24 |
| FCBF | 9 | 4 | 11 | **6** | 24 |
| FCBF (Filtered) | **1** | 2 | 9 | 14 | 25 |
| LOGO | 10 | 2 | **8** | 12 | 27 |
| LOGO (Filtered) | 7 | 1 | 11 | 11 | 28 |
| FGM | 9 | 2 | **8** | 7 | **21** |
| FGM (Filtered) | 10 | 1 | 9 | 15 | 25 |
| ours ($F_3$) | 6 | 1 | 11 | 13 | 31 |
| ours ($W_3$) | 8 | **0** | 9 | 11 | 28 |
| ours ($F_{10}$) | 6 | 1 | 11 | 12 | 28 |
| ours ($W_{10}$) | 9 | 1 | 15 | 13 | 29 |

## C.2 Accuracy and runtime results on the NIPS Datasets

The results on NIPS dataset from different methods are shown in table 4 below.

Note that the numbers we reported are runtimes *without* running tests in parallel. Since our tests are totally independent, the parallel speed up factor will be essentially linear in the partition size.

The NIPS and micro-array datasets experiments were all completed on a machine with I7-3930K 3.20GHZ 6-core CPU and 32GB RAM with 12 threads. The running time of different methods are listed in the following table 5. and 3.

# D Additional results on the large dataset

## D.1 Top-features on all relations

As we shown inf Figure 2(b), the top-features extracted by our method makes intuitive sense for relations **Spouse**, **MemberOf**, and **TopMember**. Figure 8 shows the result for other relations using the same protocol we described in the body of this paper.

We see that for most relations, the top features selected by our method makes intuitive sense, which implies the effectiveness of our approach. For relations like **per:stateorprovinces_of_residence**, the top keywords are not direct indicator of the relation (although they strongly imply the relation), this is a known problem of how the training set is generated [27, 17], and is orthogonal to the feature selection process.

Table 3: Micro-array dataset runtime performance (in seconds)

| Methods/Dataset | Colon | Leukemia | Lung | Lymph | NCI9 |
|---|---|---|---|---|---|
| MIM | 1.77 | 7.27 | 0.36 | 6.17 | 9.08 |
| MIM (Filtered) | 0.23 | 0.78 | 0.09 | 0.70 | 0.96 |
| MRMR | 10.96 | 50.36 | 2.02 | 41.72 | 57.27 |
| MRMR (Filtered) | 1.26 | 4.99 | 0.39 | 4.41 | 5.78 |
| JMI | 17.46 | 76.29 | 4.45 | 90.43 | 127.78 |
| JMI (Filtered) | 1.99 | 7.75 | 0.83 | 9.15 | 12.93 |
| ICAP | 37.80 | 167.25 | 8.73 | 180.59 | 248.40 |
| ICAP (Filtered) | 4.28 | 17.24 | 1.65 | 18.38 | 25.01 |
| DISR | 28.13 | 118.22 | 7.20 | 144.71 | 206.25 |
| DISR (Filtered) | 3.17 | 12.07 | 1.36 | 15.22 | 20.75 |
| CMIM | 1.19 | 3.47 | 1.12 | 5.49 | 2.78 |
| CMIM (Filtered) | 0.77 | 1.42 | 0.75 | 2.64 | 1.16 |
| CIFE | 37.53 | 166.53 | 8.85 | 185.69 | 259.42 |
| CIFE (Filtered) | 4.25 | 16.76 | 1.64 | 18.25 | 25.73 |
| FCBF | 14.01 | 87.17 | 34.70 | 3991.4 | 838.07 |
| FCBF (Filtered) | 2.10 | 18.76 | 5.50 | 114.24 | 158.39 |
| LOGO | 32.51 | 180.58 | 66.99 | 156.66 | 86.84 |
| LOGO (Filtered) | 14.87 | 27.53 | 52.37 | 102.49 | 53.34 |
| FGM | 1.73 | 3.30 | 4.54 | 86.71 | 142.44 |
| FGM (Filtered) | 1.15 | 1.15 | 0.83 | 5.33 | 5.61 |
| ours ($F_3$) | 1.01 | 5.35 | 0.25 | 5.01 | 6.61 |
| ours ($W_3$) | 19.72 | 82.87 | 46.21 | 1112.22 | 1599.15 |
| ours ($F_{10}$) | 2.80 | 14.79 | 0.74 | 14.50 | 19.68 |
| ours ($W_{10}$) | 66.71 | 274.12 | 153.23 | 3699.2 | 5233.84 |

Table 4: Accuracy results from different methods on NIPS datasets

| | Datasets | | | |
|---|---|---|---|---|
| | GISETTE | | MADELON | |
| Methods | BER (%) | Features (%) | BER (%) | Features (%) |
| MIM | 3.15 | 9.40 | 12.33 | 2.80 |
| MIM (Filtered) | 3.08 | 6.42 | 12.33 | 2.80 |
| MRMR | 3.69 | 8.04 | 47.83 | 9.40 |
| MRMR (Filtered) | 4.58 | 4.62 | 46.17 | 9.20 |
| JMI | 4.02 | 1.94 | 11.28 | 2.00 |
| JMI (Filtered) | 4.63 | 5.62 | 11.28 | 2.00 |
| ICAP | 4.58 | 6.24 | 12.33 | 2.80 |
| ICAP (Filtered) | 4.17 | 4.62 | 12.33 | 2.80 |
| DISR | 3.06 | 7.32 | 10.61 | 1.80 |
| DISR (Filtered) | 2.92 | 7.02 | 14.22 | 2.60 |
| CMIM | 4.46 | 3.16 | 12.33 | 2.80 |
| CMIM (Filtered) | 4.82 | 2.22 | 12.33 | 2.80 |
| CIFE | 7.82 | 9.74 | 39.83 | 10.20 |
| CIFE (Filtered) | 7.80 | 9.62 | 39.33 | 3.60 |
| FCBF | 16.86 | 0.02 | 45.50 | 0.20 |
| FCBF (Filtered) | 16.86 | 0.02 | 45.50 | 0.20 |
| LOGO | 3.09 | 4.00 | 43.94 | 17.80 |
| LOGO (Filtered) | 3.40 | 1.38 | 21.11 | 11.60 |
| FGM | **2.15** | **0.70** | 39.50 | 9.00 |
| FGM (Filtered) | 2.54 | 1.00 | 39.11 | 1.40 |
| ours ($F_3$) | 4.85 | 9.34 | 22.61 | 4.40 |
| ours ($W_3$) | 2.72 | 6.30 | **10.17** | **2.40** |
| ours ($F_{10}$) | 4.69 | 9.94 | 18.39 | 1.40 |
| ours ($W_{10}$) | 2.89 | 9.18 | 10.50 | 2.40 |

Table 5: NIPS dataset runtime performance (in seconds)

| Methods/Dataset | GISETTE | MADELON |
|---|---|---|
| MIM | 0.79 | 0.04 |
| MIM (Filtered) | 0.22 | 0.01 |
| MRMR | 23807.11 | 5.39 |
| MRMR (Filtered) | 3439.96 | 4.68 |
| JMI | 5303.93 | 7.86 |
| JMI (Filtered) | 963.97 | 2.07 |
| ICAP | 30901.43 | 59.68 |
| ICAP (Filtered) | 5866.00 | 20.08 |
| DISR | 5533.05 | 12.49 |
| DISR (Filtered) | 864.76 | 3.16 |
| CMIM | 3162.91 | 12.88 |
| CMIM (Filtered) | 3030.55 | 11.61 |
| CIFE | 30658.80 | 45.52 |
| CIFE (Filtered) | 5715.51 | 17.72 |
| FCBF | 107.57 | 0.94 |
| FCBF (Filtered) | 13.65 | 0.44 |
| LOGO | 19303.79 | 18.17 |
| LOGO (Filtered) | 2441.69 | 9.73 |
| FGM | 36.11 | 6.48 |
| FGM (Filtered) | 27.87 | 4.09 |
| ours ($F_3$) | 3.32 | 0.23 |
| ours ($W_3$) | 11.03 | 0.32 |
| ours ($F_{10}$) | 10.48 | 0.72 |
| ours ($W_{10}$) | 36.12 | 1.05 |

Figure 7: Precision/Recall on TAC-KBP with Number of Features $K = 10$ and $K = 100$.

## D.2  On varying the number selected features

Figure 2(a) shows the Precision/Recall on TAC-KBP data set with the number of selected features $K = 1000$. Figure 7 shows the result for $K = 10$ and $K = 100$. We can see that different approaches perform similarly as $K = 1000$ case.

| Relation | Keywords | | | | |
|---|---|---|---|---|---|
| org:city_of_headquarters | based | headquarters | COXnet directed | seized control of | rulers |
| org:founded_by | founder | leader | chairman | co-founder | executive |
| org:parents | employees | owned | unit | divisions | subsidiary |
| org:subsidiaries | employees articles | owned | unit | divisions | subsidiary |
| org:top_members_employees | head | executive | chairman | general | president |
| per:children | son | father | daughter | mother | said |
| per:cities_of_residence | executive | president | chairman | executive | born in |
| per:city_of_birth | born | ARodriguez | Peavy | told in | native |
| per:city_of_death | died | died home | died hospital | killed attack city | assassination |
| per:countries_of_residence | mayor | said in | born in | Democrat | told in |
| per:employee_of | executive | chairman | president | professor | director |
| per:member_of | leader | member | rebels | commander | iraq leader |
| per:parents | son | father | daughter | mother | sons |
| per:schools_attended | holds degree from | standout | graduate | student | attend |
| per:siblings | brother | sister | half-brother | found along-with | pregnancy give shops |
| per:spouse | wife | pictures | husband | married | widower |
| per:stateorprovinces_of_residence | governor of | senator from | Republican | Democrat | Republican of |

Figure 8: Top Keywords for All 17 Relations We Considered in TAC-KBP

# E  Stability Experiments

Given different data samples from the same distribution, the feature selection algorithm should ideally identify the same set of features assuming there is a unique set of "true" features[10]. However, due to biases incurred during data sampling and the redundancy present in the data, the algorithm may end up identifying different sets of features leading to inconsistency. Kuncheva [13] presented a *consistency index* which measures the consistency between two sets, with a positive value indicating similar sets, negative value for anti-correlation and zero for random relations.

For measuring the consistency index, we take 50 bootstraps from a dataset and select feature on the bootstraps. The consistency index of the dataset from a particular method is taken as the median value from the 50 boot-straps. The box plot of consistency index from different methods on the 15 UCI datasets are shown in figure 9. In general, filter method has relatively higher stability as compared to wrapper method. The stability measure

Figure 9: Consistency index across 15 UCI datasets.

of the filter method is very similar to JMI and MIM method, which is attributed to the similarity in obtaining the scores. From the stability measure of our method in figure 9, we can also observe that as we increase the number of tests, the algorithm gets more stable, which confirms the experiments we did in previous section.

## Footnotes

[10]This will not hold in case there are multiple subsets of features that are equally good.