[Reviews · NeurIPS 2014]

Submitted by Assigned_Reviewer_12

In this paper a novel and interesting parallel feature selection framework based on group testing is proposed for large scale data. As the author claimed, the presented method can speed up the feature selection algorithm and provide superior performance than other existing methods especially on very high dimensional dataset.

The proposed framework for parallel feature selection is well defined with sufficient theoretical analysis. The author has proved that KL divergence and MI is C-separable under certain conditions. But for real application, it lacks good description in detail for the feature selection algorithm in parallel form. How to employ group testing to parallelize the proposed or existing feature selection algorithms? Some algorithm definition or pseudo-code is necessary for the readers’ better understandings.

As stated in subsection 3.2.2, $F_k$ denotes filter method with MI score functions and $W_k$ denotes wrapper method with logistic regression. While how to achieve these algorithms is not discussed including the parameters setting, etc. In addition, it needs to add a conclusion part for this paper to summarize the proposed method and experiments.
Summary: This paper presents a novel framework for parallel feature selection based on group testing. Parallel feature selection algorithm in detail is not well explained. It is a good paper with potential applications.

Submitted by Assigned_Reviewer_27

This paper proposes a novel method for feature selection based on independent group testing, which is desirable for parallel execution for feature selection. The method consists of three steps: (1) test design, (2) score function, and (3) feature identification. In test design, groups of features for being tested are determined, score function assigns a score to each group, and feature identification step finally selects a set of features based on the scores. This paper presented a theory including the number of tests needed to maintain feature selection accuracy, and the experimental results which support that the proposed approach outperforms the competitors in terms of accuracy, scalability and robustness.

Major comments:

1. In line 132, groups were randomly generated. Are there advantages or disadvantages of making small or large groups?
In test design, it would be useful if authors include more discussion on how groups of features should be made both in theory and practice.

2. In experiments, as competing methods, popular sparse regression based feature selection methods should be included (e.g. Lasso or L1-logistic regression).

3. In Figure 2, why does the proposed method improve the accuracy of FGM but not MIM when used as preprocessing step?

4. It would be good to add more synthetic experiments to verify that theory is in agreement with the experiments.

5. Conclusion or discussion section is missing.

Minor comments:

1. In line 82, there a typo. schem -> scheme

2. Section 2 and 3 can be structured for better readability.

3. In Figure 1, x-axis label is missing.

4. In Figure 2, two dotted lines overlap, and it is hard to distinguish between the two lines.
Summary: Overall, the proposed method looks interesting, and this will be a useful direction for scalable feature selection. Writing can be further improved for better readability.

Submitted by Assigned_Reviewer_40

The paper proposed a parallel feature selection for supervised learning. Idea is to obtain multiple subsets of the full feature select and evaluate features’ score on each feature subset in parallel. This idea does not sounds new and have a potential problem that links to scalability. Specifically, this parallel approach request each node of the grid system hold the whole data for sampling features. This will significantly affect the scalability of the algorithm when the size of the data is large, since it might be impossible to put all data on one node. A popular way to handle large data set is data partition based on sample. However, the proposed method is not suitable for this data partition strategy. In the paper the authors also need to provide a time complexity analysis as well as a scalability analysis that consider communication cost in a distributed computing environment.
Summary: The idea presented in the paper does not sounds new and have a potential problem that links to scalability.
Author Feedback
Author rebuttal: Reviewer #1:
But for real application, it lacks good description in detail for the feature selection algorithm in parallel form. How to employ group testing to parallelize the proposed or existing feature selection algorithms? Some algorithm definition or pseudo-code is necessary for the readers’ better understandings.

We agree that our writing can be improved. To parallelize the method: each node select its own random tests, performs the tests, and then we centrally collect the scores. The central node does the final ranking. Due to the fact that all tests are random and independent, no coordination is necessary until the final dot-products are communicated. The communication cost is essentially the dot-products, which is small.

To be more precise, here is how we can parallelize the method. Recall that the test matrix A is selected at random, each row (in fact each entry) is selected randomly. Let’s say we have k parallel nodes. Then, each node can be responsible for 1/k the total number of rows. Essentially, we vertically partition A into k layers, each layer has t/k rows. Each parallel node is responsible for constructing its layer and collecting the test scores corresponding to that layer. To obtain the rank of the j’th feature, we need to compute the dot-product < a_j, s >, which is the sum of k dot-products, one for each layer.

As stated in subsection 3.2.2, $F_k$ denotes filter method with MI score functions and $W_k$ denotes wrapper method with logistic regression. While how to achieve these algorithms is not discussed including the parameters setting, etc. In addition, it needs to add a conclusion part for this paper to summarize the proposed method and experiments.

We quantize the data before doing MI calculation, and the probabilities for MI are all estimated by maximum likelihood. For wrapper method we use standard logistic regression.

Reviewer #2:
1. In line 132, groups were randomly generated. Are there advantages or disadvantages of making small or large groups?

Yes, the advantage of having large groups (up to N/2) is that theoretical that can make the total number of tests smaller (down to O(\log N)). The disadvantage of having large groups is that the scores are harder to estimate correctly.

In test design, it would be useful if authors include more discussion on how groups of features should be made both in theory and practice.

Theorem 2.2 gives a theoretical guide to setting these parameters, but it is not optimal. In practice, it is necessary to set p to be a little small so that we can obtain reasonable estimate of scores from each test.

2. In experiments, as competing methods, popular sparse regression based feature selection methods should be included (e.g. Lasso or L1-logistic regression).

We have included the feature generating machine (FGM), which is a highly competing method proposed recently that builds on L1- based regularization. We have tried lasso regression on most of the datasets, and did not obtain good results, and hence was not included.

3. In Figure 2, why does the proposed method improve the accuracy of FGM but not MIM when used as preprocessing step?

MIM is a special case from our proposed method, where one set the test matrix to be a NxN identity matrix (N is the data dimension). One possible reason is that the two schemes are so similar so we do not get much information gain by running ours as a preprocessing, but the scheme between FGM and ours are significantly different and thus providing more benefits by running ours as a preprocessing step. However, we do get about 20% speed up from MIM by running our method first.

4. It would be good to add more synthetic experiments to verify that theory is in agreement with the experiments.

We have a set of synthetic experiments that verifies the theory, but because of space constraint we have put them in the supplementary materials in section B. We do plan to include a summary of these results in the paper.

Reviewer #3:
This idea does not sounds new and have a potential problem that links to scalability.

We would appreciate it very much if the reviewer can point us to a reference where this method was proposed. We are not aware of any existing feature selection method that is the same as ours.

Specifically, this parallel approach request each node of the grid system hold the whole data for sampling features. This will significantly affect the scalability of the algorithm when the size of the data is large, since it might be impossible to put all data on one node. A popular way to handle large data set is data partition based on sample. However, the proposed method is not suitable for this data partition strategy. In the paper the authors also need to provide a time complexity analysis as well as a scalability analysis that consider communication cost in a distributed computing environment.

Perhaps it was not clear, but our method can be directly used with random data partitioning.

First, all we need at each site is to construct its own random tests, score them, compute the dot-products, and the only thing that has to be communicated to some central site is the final dot-product. The dot-product decomposes into a sum of dot-products (a vertical partition of the test matrix).

Second, with random data partitions, scores can be estimated from the samples. The more random the data in the partition strategy, the better the method performs. In particular, the whole data need not ever be on a single machine for this to test to be performed.

Third, the fact that all our tests are random allows for all sites to operate independently until the very last step when the dot-products are communicated, one dot-product from each site. This communication cost is very light, as we have described in our response to reviewer #1.